# Reaction of Pyrrolobenzothiazines with Schiff Bases and Carbodiimides: Approach to Angular 6/5/5/5-Tetracyclic Spiroheterocycles

**DOI:** 10.3390/molecules29092089

**Published:** 2024-05-01

**Authors:** Ekaterina A. Lystsova, Anastasia D. Novokshonova, Pavel V. Khramtsov, Alexander S. Novikov, Maksim V. Dmitriev, Andrey N. Maslivets, Ekaterina E. Khramtsova

**Affiliations:** 1Department of Chemistry, Perm State University, ul. Bukireva, 15, 614990 Perm, Russia; liscova_ea@mail.ru (E.A.L.); maxperm@yandex.ru (M.V.D.); koh2@psu.ru (A.N.M.); 2Department of Biology, Perm State University, ul. Bukireva, 15, 614990 Perm, Russia; anast218bio@gmail.com (A.D.N.); khramtsovpavel@yandex.ru (P.V.K.); 3Institute of Ecology and Genetics of Microorganisms, Perm Federal Research Center, The Ural Branch of Russian Academy of Sciences, ul. Goleva, 13, 614081 Perm, Russia; 4Institute of Chemistry, Saint Petersburg State University, Universitetskaya nab. 7/9, 199034 St. Petersburg, Russia; a.s.novikov@spbu.ru; 5Research Institute of Chemistry, Peoples’ Friendship University of Russia (RUDN University), ul. Miklukho-Maklaya, 6, 117198 Moscow, Russia

**Keywords:** carbodiimide, *Chlorella*, DFT calculations, nitrogen heterocycle, 1*H*-pyrrole-2,3-dione, Schiff base, sulfur heterocycle

## Abstract

1*H*-Pyrrole-2,3-diones, fused at [*e*]-side with a heterocycle, are suitable platforms for the synthesis of various angular polycyclic alkaloid-like spiroheterocycles. Recently discovered sulfur-containing [*e*]-fused 1*H*-pyrrole-2,3-diones (aroylpyrrolobenzothiazinetriones) tend to exhibit unusual reactivity. Based on these peculiar representatives of [*e*]-fused 1*H*-pyrrole-2,3-diones, we have developed an approach to an unprecedented 6/5/5/5-tetracyclic alkaloid-like spiroheterocyclic system of benzo[*d*]pyrrolo[3′,4′:2,3]pyrrolo[2,1-*b*]thiazole via their reaction with Schiff bases and carbodiimides. The experimental results have been supplemented with DFT computational studies. The synthesized alkaloid-like 6/5/5/5-tetracyclic compounds have been tested for their biotechnological potential as growth stimulants in the green algae *Chlorella vulgaris*.

## 1. Introduction

To improve the clinical success, reduce the undesirable side effects caused by the binding promiscuity of drug candidates, and speed up the lead optimization process, it is necessary to look for ways to expand the medicinal chemistry synthetic toolbox to be able to target more complex three-dimensional (3D) chemical space [1,2,3,4]. The 3D shape of a molecule is the most important factor determining its biological activity [5,6,7]. Due to these, angular polycyclic alkaloid-like spirocycles are attractive objects for drug discovery and related studies [8].

[*e*]-Fused 1*H*-pyrrole-2,3-diones (FPDs) (Figure 1) are versatile starting materials for the synthesis of various heterocyclic systems [9,10,11,12], including angular polycyclic alkaloid-like spiroheterocycles (for example, 6/6/5/5- [13], 6/6/5/6- [14], 6/6/5/6/6- [15], 6/6/5/6/5- [15], 6/5/7/5- [16], 6/5/7/6- [17], 6/6/5/7/6- [18], 6/7/5/6- [19], 5/6/5/6-systems [20] and some others (Figure 1)).

Exploring the scope of the recently discovered by us nucleophile-induced ring contraction reaction in FPDs [21], we unexpectedly found an approach to an unprecedented angular 6/5/5/5-tetracyclic alkaloid-like spiroheterocyclic system of benzo[*d*]pyrrolo[3′,4′:2,3]pyrrolo[2,1-*b*]thiazole (Figure 2). Such a 6/5/5/5-tetracyclic framework is a quite interesting one, since it is present in natural products (retigeranic acid, a sesterterpene from Himalayan lichens *Lobaria retigera* and *Lobaria subretigeria* [22]) and synthetic biologically active molecules [23,24] (Figure 2).

Thus, herein, we report the first synthetic approach to an unprecedented 6/5/5/5-tetracyclic alkaloid-like spiroheterocyclic system of benzo[*d*]pyrrolo[3′,4′:2,3]pyrrolo[2,1-*b*]thiazole (Figure 2) via the reaction of pyrrolobenzothiazines (FPDs incorporating 1,4-benzothiazine moiety) with Schiff bases and carbodiimides. The experimental results were supplemented with DFT computational studies to elucidate the mechanism and stereoselectivity of the reaction. The biotechnological potential of the reported benzo[*d*]pyrrolo[3′,4′:2,3]pyrrolo[2,1-*b*]thiazoles as growth stimulants and promoters of pigments accumulation in the green algae *Chlorella vulgaris* was demonstrated.

## 2. Results and Discussion

### 2.1. Chemistry

Recently, we reported [25] a new class of FPDs, aroylpyrrolobenzothiazinetriones (APBTTs) **1** (Figure 1). APBTTs **1** were found to react as oxadienes in a hetero-Diels–Alder reaction with electron-rich dienophiles (alkoxyolefins, styrene) (Figure 1) [25], which is a quite common reactivity for other known types of FPDs and monocyclic 1*H*-pyrrole-2,3-diones [9,10,14,15]. However, under the action of mononucleophiles (amines and alcohols), APBTTs **1** were found to undergo a ring-contraction reaction (Figure 1) [21], which greatly distinguished their reactivity from the reactivity of their 5-oxa- and 5-aza-analogues [10].

5-Oxa- and 5-aza-analogues of APBTTs **1** seem not to react with Schiff bases and carbodiimides (C=N reagents) without thermal decomposition (for such reactions under thermal decomposition conditions, see the work presented in [12]). At the same time, monocyclic 1*H*-pyrrole-2,3-diones are known to react with carbodiimides as oxadienes in formal hetero-Diels–Alder reactions to produce the corresponding cycloadducts (Figure 2) [26].

Considering the tendency for the unusual reactivity of APBTTs **1**, we studied their reaction with Schiff bases and carbodiimides under conditions without thermal decomposition of APBTTs **1** (for thermal decomposition of APBTTs **1**, see the work presented in [27]).

To start with, we tested the reaction of APBTT **1a** with *N*-benzylideneaniline **2a** (Figure 3). As a result, 6/5/5/5-tetracyclic product **3a** was isolated by a simple crystallization from the reaction mixture in the yield of 52% (Figure 3). Compound **3a** was obtained as a single (3*R**,3a*S**,11a*R**)-diastereomer, and its structure was unequivocally determined by a single crystal X-ray analysis (CCDC 2341688).

Obviously, our hypothesis of the unusual reactivity of APBTTs **1** in reactions with C=N reagents was confirmed, which justified a more in-depth study of this transformation.

Next, we carried out a series of experiments to optimize the conditions of the reaction of APBTT **1a** with Schiff base **2a** (Table 1).

According to Table 1, APBTT **1a** and Schiff base **2a** reacted most quickly in polar solvents (acetone, DMAA, DMSO, DMF, NMP (entries 1, 7–10, Table 1)), but the reaction proceeded unselectively (a difficult to identify mixture of products was observed), and only trace amounts of the target product **3a** were formed. Interestingly, in acetonitrile (entry 2, Table 1), which is a polar solvent too, the reaction yield was much higher, which could indicate that the low yields of the product **3a** in polar solvents were caused not only by the polarity of these solvents, but also by their specific solvation effects and their ability to react with the reaction intermediates. In nonpolar solvents (entries 3–6, 12, 13 Table 1), the reaction proceeded much slower and was more selective towards the product **3a**. The best yield of the product **3a** (HPLC yield of 90%) was observed in chloroform when heated for 5 h (entry 5, Table 1).

It should be mentioned that, during the optimization of the reaction of APBTT **1a** with the Schiff base **2a** at room temperature in anhydrous butyl acetate, acetonitrile, and acetone, we observed the formation of product **4a** in significant amounts (Figure 4). Under these conditions, the reaction proceeded very slowly (about a month), and obviously, side reactions took place. Since the reaction vials were not sealed, it could be assumed that atmospheric moisture affected the reaction (Figure 4). Moreover, the reaction of APBTT **1a** with the Schiff base **2a** at room temperature in acetic acid, containing traces of water, produced compound **4a** in 24 h in a very good isolated yield (86%) (Figure 4).

To prove our hypothesis of the formation of compound **4a**, we carried out a reaction of APBTTs **1a**,**b** with aniline **5a** and benzylamine **5b** (Figure 5). As a result, we isolated target products **4a**,**b** in yields of 95% and 46%, respectively. The structure of compound **4b** was unequivocally confirmed by a single crystal X-ray analysis (CCDC 2341690).

Then, we performed reactions of APBTTs **1a**–**f** with Schiff bases **2a**–**g** and azine **2h** to determine the reactant scope (Table 2).

As a result, we found that the reaction of APBTTs **1** with Schiff bases **2** performed under optimized conditions (chloroform as the solvent) produced target products **3** in poor to very good HPLC yields (Conditions A, Table 2). However, our attempts to isolate products **3** from such reaction mixtures (in scale of 298 μmol) were unsuccessful. Moreover, we observed that compounds **3** underwent unfavorable transformations during our attempts to isolate and purify them by column chromatography. For these reasons, we replaced chloroform with benzene in these reactions. As a result, we noticed a decreasing tendency in the HPLC yields of products **3** (Conditions B, Table 2), which correlated with the optimization data for the test reaction of APBTT **1a** with Schiff base **2a** (Table 1). However, products **3** were easily isolated and purified by simple crystallization directly from the reaction mixtures (in scale of 298 μmol, benzene).

We also observed that the nature of the aryl substituents in the examined (Table 2) APBTTs **1** and Schiff bases **2** did not significantly affect the yields of the corresponding products **3**. However, the reactions of APBTT **1a** with 1-phenyl-*N*-(pyridin-2-yl)methanimine 2**i**, 1-((phenylimino)methyl)naphthalen-2-ol 2**j**, *N*-(4-nitrophenyl)-1-phenylmethanimine, *N*-mesityl-1-(4-nitrophenyl)methanimine 2**k**, *N*-(2-chlorophenyl)-1-phenylmethanimine 2**l**, and *N*,*N*-dimethyl-4-((phenylimino)methyl)aniline 2**m** did not produce the desired products **3**, which was possibly caused by the presence of additional nucleophilic centers, *o*-substituents, or strong electron-withdrawing groups in the molecules of these Schiff bases. Moreover, the reaction with *N*-benzylmethanimine 2**n** did not give the corresponding product **3** too, possibly due to the absence of aromatic substituent at the CH part of this Schiff base, which could stabilize reaction intermediates.

Noteworthy, when synthesizing product **3d**, we succeeded in isolating by-product **6d** from the reaction mixture in the isolated yield of 2% (Figure 6). The structure of compound **6d** was unequivocally confirmed by a single crystal X-ray analysis (CCDC 2341691). Apparently, the formation of product **6d** proceeded through an intermediate **A** (conditions for formation of compounds **A** were discussed in the work presented in [21]) and the hydrolysis of Schiff base **2a** (Figure 6).

Moreover, in cases of compounds **3j** and **3k**, we succeeded in isolating their stereoisomers, compounds **3′j** and **3′k** (isolated yields of 7 and 28%, respectively). Indeed, for structures of compounds **3**, there are four possible diastereomers with (3*R**,3a*S**,11a*R**), (3*S**,3a*S**,11a*R**), (3S*,3a*R**,11a*R**), and (3*R**,3a*R**,11a*R**) relative configurations (Figure 3). In our scope studies (Table 2), we isolated exclusively (3*R**,3a*S**,11a*R**) diastereomers for compounds **3a**–**i**,**l**,**m**. In our scope studies (Table 2), we were unable to detect and determine any other diastereomers by HPLC due to the absence of their reference samples; NMR spectra of crude reaction mixtures were also not suitable for these purposes. But, in cases of compounds **3j** and **3k**, we isolated both (3*R**,3a*S**,11a*R**) diastereomers **3j**,**k** and their stereoisomers **3′j**,**k** with unknown relative configurations. Diastereomers **3′j**,**k** were found to be unstable in HPLC studies, long NMR experiments in solutions and during our attempts to grow a crystal suitable for a single crystal X-ray analysis, which was possibly caused by their reaction with water.

The ^1^H NMR and IR spectra of compounds **3j**,**k** and compounds **3′j**,**k** are very similar and cannot be used for their identification relative to each other. But their ^13^C NMR spectra are quite different in the region of 62–82 ppm (where sp^3^ atoms—C*^3^*, C*^3a^*, C*^11a^*—appear), which is enough to distinguish them from each other. Fragments of their ^13^C NMR spectra in these regions look as follows:^13^C NMR of **3j** (100 MHz, DMSO-*d*_6_): δ = 79.3, **66.0**, 65.2 ppm;
^13^C NMR of **3′j** (100 MHz, DMSO-*d*_6_): δ = 79.2, **70.6**, 64.2 ppm;
^13^C NMR of **3k** (100 MHz, DMSO-*d*_6_): δ = 79.6, **65.7**, 65.5 ppm;
^13^C NMR of **3′k** (100 MHz, DMSO-*d*_6_): δ = 79.3, **70.8**, 64.5 ppm.

However, this information is not enough to determine the relative configuration of compounds **3′**.

Then, to elucidate the possible mechanism of the reaction and compare the thermodynamics and kinetics of possible stereoisomers formation, we performed computational DFT studies of a model reaction between the APBTT **1a** and Schiff base **2a** (Figure 7). We proposed several reaction pathways for **1a** → **3a** transformation (Figure 7), but the results of DFT calculations revealed only one, very energetically unprofitable, intermediate **I2** on the potential energy surface and indicated that the formation of product **3a** occurred directly from the orientation complex **OC**. The hypothetical transformation **OC** → **3a** (via transition state **TS** (Figure 4)) is less thermodynamically profitable (by 4.7 kcal/mol in terms of Gibbs-free energies of reaction, Table 3, Figure 5) compared to the alternative hypothetical transformation **OC** → **3′a** (via transition state **TS’** (Figure 4)) but more kinetically favorable (by 1.6 kcal/mol in terms of Gibbs-free energies of activation, Table 3, Figure 5). Thus, diastereomer **3a** is a kinetically controlled product, and diastereomer **3′a** is a thermodynamically controlled one (Figure 5).

Since our computational studies revealed only intermediate-**I2**-like transition states **TS** and **TS′** (their 3D structures are available in Appendix A) in the reaction, and these transition states could afford only diastereomers (3*R**,3a*S**,11a*R**) and (3*S**,3a*S**,11a*R**) (Figure 7), we suggest that compounds **3′j**,**k** had (3*S**,3a*S**,11a*R**) a relative configuration (Figure 3).

Then, in order to expand the reagent scope of the reaction, we studied the reactions of APBTTs **1** with carbodiimides **7**.

First, we tested the reaction of APBTT **1a** with *N*,*N*′-dicyclohexylcarbodiimide (DCC) **7a** (Figure 8). As we expected, 6/5/5/5-tetracyclic product **8a** was isolated by a simple crystallization from the reaction mixture in the yield of 44% (Figure 8). Compound **8a** was obtained as a single diastereomer, and its structure was unequivocally determined by a single crystal X-ray analysis (CCDC 2341689).

Next, we performed the optimization of the reaction conditions (Table 4). As a result, we did not observe any correlation between the polarity of the solvent and the yield of the product **8a**. The best yield of the product **8a** (HPLC yield of 96%) was observed in 1,4-dioxane when heated for 40 min (entry 6, Table 4). Our optimization studies at room temperature, as expected, revealed that reactions took place over 14 days, during which time both APBTT **1a** and DCC **7a** underwent side reactions with water.

Then, we performed reactions of APBTTs **1a**–**g** with carbodiimides **7a**,**b** to determine the reactant scope (Table 5). Initially, we tried to perform the scope examination under optimal conditions in 1,4-dioxane (entry 6, Table 4). But under these conditions, our attempts to isolate products **8** from such reaction mixtures (in scale of 298 μmol) were unsuccessful, and we observed that compounds **8** underwent unfavorable transformations during our attempts to isolate and purify them by column chromatography. A similar situation was observed when we tried to apply acetonitrile as the reaction solvent (entry 2, Table 4). Thus, we had to carry out these reactions in toluene (entry 11, Table 4). In toluene, products **8** were easily isolated by simple crystallization from the reaction mixtures.

As a result, we found that the reaction of APBTTs **1** with carbodiimides **7** performed in toluene produced target products **8** in fair to good HPLC yields (Table 5). We observed that the nature of the aryl substituents in examined (Table 5) APBTTs **1** did not significantly affect the yields of the corresponding products **8**.

### 2.2. Biology

The green unicellular algae *Chlorella vulgaris* serves as a source of valuable metabolites for the food industry [28,29,30], agriculture [22,23,24,25,26,27,28,29,30,31,32,33,34], cosmetics [35,36,37], and biodiesel production [38,39,40,41,42]. There is a significant demand for chemical stimulants that promote the growth of this algae, as well as the accumulation of lipids [43,44,45], proteins [46,47], carbohydrates [48,49,50], and pigments [51] such as chlorophylls [52,53] and carotenoids [54,55,56,57,58]. We selected several synthesized compounds with favorable characteristics, such as high synthesis yield and improved solubility in polar solvents. These compounds were added in varying concentrations to *C. vulgaris* cultures, which were then cultivated, followed by the measurement of algae growth (cell concentration) and the accumulation of pigments.

The bioactivity study was conducted in two stages. Initially, a screening experiment in small volumes of algal cultures (96-well plates) was performed (Table 6). Two compounds (**3a** and **8j**) that promoted the growth of *C. vulgaris* were further analyzed in a subsequent experiment. Algae were grown in 50-mL Erlenmeyer flasks in the presence of the tested compounds at concentrations ranging from 1 × 10^−7^ mol/L to 1 × 10^−4^ mol/L. Glucose served as a positive control due to its ability to enhance algae growth, while the negative control was culture fluid containing 1% DMSO, which was used for the dilution of the tested compounds.

Both **3a** and **8j** increased the chlorophyll content in cells and/or algae growth at specific concentrations (Table 7 and Table 8). The most notable effect was the nearly 30% increase in chlorophyll content in cells in the presence of 1 × 10^−4^ mol/L **3a**, although this was accompanied by a 17.8% decrease in cell growth. Chlorophylls are utilized as natural colorants in the food, cosmetics, and textile industries [59]. As *C. vulgaris* cells are enriched with chlorophyll (up to 4.5% of dry weight), it can be considered one of the most prominent natural sources of these pigments with substantial commercial potential [60]. Therefore, the synthesized compounds can be utilized to enhance the efficiency of algal chlorophyll production for industrial applications.

## 3. Materials and Methods

### 3.1. Synthetic Methods and Analytic Data of Compounds

#### 3.1.1. General Information

^1^H and ^13^C NMR spectra (Appendix A) were acquired on a Bruker Avance III 400 HD spectrometer (Bruker BioSpin AG, Faellanden, Switzerland) (at 400 and 100 MHz, respectively) in CDCl_3_ or DMSO-*d_6_*, using solvent residual signals (in ^13^C NMR, 77.00 for CDCl_3_, 39.52 for DMSO-*d_6_*; in ^1^H NMR, 7.26 for CDCl_3_, 2.50 for DMSO-*d_6_*) as internal standards. ^19^F NMR spectra (Appendix A) were acquired on a Bruker Avance III 400 HD spectrometer (Bruker BioSpin AG, Faellanden, Switzerland) (at 376 MHz) in CDCl_3_ or DMSO-*d_6_* using no internal standard. IR spectra were recorded on a Perkin–Elmer Spectrum Two spectrometer (PerkinElmer Inc., Waltham, MA, USA) from mulls in mineral oil. Melting points were measured on a Mettler Toledo MP70 apparatus (Mettler-Toledo (MTADA), Schwerzenbach, Switzerland). Elemental analyses were carried out on a Vario MICRO Cube analyzer (Elementar Analysensysteme GmbH, Langenselbold, Germany). The reaction conditions were optimized using HPLC-UV on Hitachi Chromaster (Hitachi High-Tech, Tokyo, Japan) [NUCLEODUR C18 Gravity column (particle size 3 μm; eluent acetonitrile–water, flow rate 1.5 mL/min); Hitachi Chromaster 5430 diode array detector (λ 210–750 nm)]. The single crystal X-ray analyses of compounds **3a**, **4b**, **6d**, **8a** were performed on an Xcalibur Ruby diffractometer (Agilent Technologies, Wroclaw, Poland). The empirical absorption correction was introduced by a multi-scan method using the SCALE3 ABSPACK algorithm [61]. Using OLEX2 [62], the structures were solved with the SHELXT [63] or SUPERFLIP [64] program and refined by the full-matrix least-squares minimization in the anisotropic approximation for all non-hydrogen atoms with the SHELXL [65] program. Hydrogen atoms bound to carbon were positioned geometrically and refined using a riding model. The hydrogen atoms of NH and OH groups were refined independently with isotropic displacement parameters. The APBTTs **1a**–**g** were obtained according to reported procedures [21,25]. The compounds **2a**–**n** were obtained according to the reported procedures [66,67,68,69,70,71,72,73,74,75,76,77,78,79,80]. Benzene, toluene, *o*-xylene, *p*-xylene, 1,4-dioxane, and THF for procedures with compounds **1** were distilled over Na before use. Acetone, butyl acetate, and chloroform for procedures with compounds **1** were distilled over P_2_O_5_ before the use. DMAA, DMF, DMSO, NMP, and acetonitrile for procedures with compounds **1** were dried over molecular sieves 4Å before the use. All procedures with APBTTs **1** were performed in an oven-dried glassware. All other solvents and reagents were purchased from commercial vendors and were used as received. Thin-layer chromatography (TLC) was performed on ALUGRAM Xtra SIL G/UV254 silica gel 60 plates (Macherey-Nagel, Düren, Germany) using EtOAC/toluene, 1:5 *v*/*v*, EtOAc, toluene as eluents; spots were visualized with iodine vapor and/or UV light (254, 365 nm) in the light of a TLC viewing cabinet Petrolaser TLC-254/365 Thin Layer Chromatography Dark Room (Petrolaser, St. Petersburg, Russia). In ^13^C NMR spectra of compounds **3a**,**c**–**g**,**i**, **3′j**, **8d**–**f**,**i**,**j**, signals of some aromatic carbons could not be found.

#### 3.1.2. Procedure to Compounds **3a**–**m**

A mixture of APBTT **1** (0.298 mmol) and Schiff base **2** (0.298 mmol) in benzene (5 mL) was heated for 3–24 h at 85 °C (until the dark violet color characteristic of APBTT **1** disappeared and a transparent yellow solution formed). Then, the reaction mixture was cooled to room temperature.

*(3R*,3aS*,11aR*)-3a-Benzoyl-2,3-diphenyl-3,3a-dihydrobenzo[d]pyrrolo[3′,4′:2,3]pyrrolo[2,1-b]thiazole-1,4,5(2H)-trione* (**3a**). The formed precipitate was filtered off. Then, the precipitate was stirred for 30 min at 40–45 °C in a mixture of toluene and ethanol (6:1 *v*/*v*, 3 mL). After that, the precipitate was filtered off and washed with a small amount of toluene (1 mL) and ethanol (1 mL) to produce compound **3a**. Yield: 80.1 mg (52%); yellow solid; mp 133–135 °C. ^1^H NMR (400 MHz, DMSO-*d_6_*): δ = 7.84 (m, 1H), 7.62 (m, 1H), 7.54–7.47 (m, 7H), 7.37 (m, 1H), 7.34 (m, 2H), 7.29 (m, 2H), 7.23 (m, 4H), 7.13 (m, 1H), 6.65 (s, 1H) ppm. ^13^C NMR (100 MHz, DMSO-*d_6_*): δ = 192.8, 191.3, 165.1, 155.6, 136.2, 135.4, 134.9, 134.0, 133.7, 130.3, 128.9 (2C), 128.7 (2C), 128.6 (2C), 128.5 (2C), 128.3 (2C), 128.3, 126.5, 126.2, 124.4 (2C), 123.1, 117.4, 79.5, 66.0, 65.5 ppm. IR (mineral oil): 1763, 1716, 1678 cm^−1^. Anal. Calcd (%) for C_31_H_20_N_2_O_4_S: C 72.08; H 3.90; N 5.42. Found: C 72.23; H 3.98; N 5.43.*(3R*,3aS*,11aR*)-3a-(4-Methylbenzoyl)-2,3-diphenyl-3,3a-dihydrobenzo[d]pyrrolo[3′,4′:2,3]pyrrolo[2,1-b]thiazole-1,4,5(2H)-trione (**3b**)*. The solvent was evaporated to 1 mL. The resulting precipitate was filtered off, washed with benzene (0.5 mL), and recrystallized from benzene (2 mL) to produce compound **3b**. Yield: 28 mg (18%); yellow solid; mp 180–182 °C. ^1^H NMR (400 MHz, DMSO-*d*_6_): δ = 7.85 (m, 1 H), 7.52 (m, 3H), 7.43 (m, 2H), 7.34 (m, 2H), 7.30 (m, 3H), 7.27–7.17 (m, 6H), 7.13 (m, 1H), 6.67 (s, 1H), 2.33 (s, 3H) ppm. ^13^C NMR (100 MHz, DMSO-*d*_6_): δ = 191.7, 191.4, 170.2, 165.1, 155.6, 144.7, 136.2, 134.8, 134.0, 132.5, 130.3, 129.5 (2C), 128.9 (2C), 128.7, 128.6 (2C), 128.5 (2C), 128.3, 126.5, 126.3, 124.4 (2C), 123.2, 117.4, 114.5, 79.4, 65.9, 65.4, 21.0 ppm. IR (mineral oil): 1785, 1716, 1672 cm^−1^. Anal. Calcd (%) for C_32_H_22_N_2_O_4_S: C 72.44; H 4.18; N 5.28. Found: C 72.67; H 4.28; N 5.32.*(3R*,3aS*,11aR*)-3a-(4-Fluorobenzoyl)-2,3-diphenyl-3,3a-dihydrobenzo[d]pyrrolo[3′,4′:2,3]pyrrolo[2,1-b]thiazole-1,4,5(2H)-trione* (**3c**). The solvent was evaporated to 2.5 mL. The obtained mixture was frozen. The resulting precipitate was filtered off, washed with benzene (0.5 mL), and recrystallized from toluene (2 mL) to produce compound **3c**. Yield: 40 mg (25%); yellow solid; mp 111–113 °C. ^1^H NMR (400 MHz, DMSO-*d*_6_): δ = 7.82 (m, 1H), 7.67 (m, 2H), 7.52 (m, 3H), 7.39 (m, 2H), 7.33 (m, 2H), 7.29 (m, 1H), 7.25 (m, 4H), 7.22 (m, 2H), 7.18–7.11 (m, 3H) ppm. ^13^C NMR (100 MHz, DMSO-*d*_6_): δ = 191.6, 190.7, 165.2, 164.9 (d, *J* = 254.5 Hz), 155.8, 136.2, 134.8, 134.1, 132.0 (d, *J* = 10.1 Hz, 2C), 131.8 (d, *J* = 3.0 Hz), 130.7, 128.6 (2C), 128.6 (2C), 126.6, 126.1, 125.3, 124.4 (2C), 123.0, 117.6, 116.0 (d, *J* = 22.2 Hz, 2C), 79.9, 65.7, 65.5 ppm. ^19^F NMR (376 MHz, DMSO-*d*_6_): δ = −104.24 ppm. IR (mineral oil): 1735, 1697, 1658 cm^−1^. Anal. Calcd (%) for 2C_31_H_19_FN_2_O_4_S·C_7_H_8_: C 71.37; H 3.99; N 4.82. Found: C 71.51; H 4.08; N 4.99.*(3R*,3aS*,11aR*)-3a-(4-Bromobenzoyl)-2,3-diphenyl-3,3a-dihydrobenzo[d]pyrrolo[3′,4′:2,3]pyrrolo[2,1-b]thiazole-1,4,5(2H)-trione* (**3d**). The solvent was evaporated to 2.5 mL, the reaction mass was frozen. The resulting precipitate was filtered off, washed with benzene (0.5 mL), and recrystallized from toluene (2 mL) to produce compound **3d**. Yield: 30 mg (17%); yellow solid; mp 207–209 °C. ^1^H NMR (400 MHz, DMSO-*d*_6_): δ = 7.81 (m, 1H), 7.75 (m, 2H), 7.53 (m, 5H), 7.33 (m, 1H), 7.29 (m, 2H), 7.26 (m, 4H), 7.22 (m, 2H), 7.13 (m, 1H), 6.57 (s, 1H) ppm. ^13^C NMR (100 MHz, DMSO-*d*_6_): δ = 192.2, 190.5, 165.08, 155.6, 136.1, 134.8, 134.2, 133.9, 131.8 (2C), 130.6 (2C), 128.8, 128.6 (2C), 128.5 (2C), 128.1, 127.7, 126.4, 126.0, 124.2 (2C), 122.8, 117.5, 79.8, 65.7, 65.4 ppm. IR (mineral oil): 1757, 1728, 1701 cm^−1^. Anal. Calcd (%) for C_31_H_19_BrN_2_O_4_S: C 62.53; H 3.22; N 4.70. Found: C 62.64; H 3.35; N 4.60.*(3R*,3aS*,11aR*)-3a-(Furan-2-carbonyl)-2,3-diphenyl-3,3a-dihydrobenzo[d]pyrrolo[3′,4′:2,3]pyrrolo[2,1-b]thiazole-1,4,5(2H)-trione* (**3e**). The resulting precipitate was filtered off, washed with benzene (1 mL) and recrystallized from toluene (2–3 mL). Then, the obtained crystals were stirred in a mixture of toluene and ethanol (5:1 *v*/*v*, 3 mL) at 50 °C for 10 min. Then, the precipitate was filtered off to produce compound **3e**. Yield: 71 mg (47%); yellow solid; mp 147–149 °C. ^1^H NMR (400 MHz, DMSO-*d*_6_): δ = 7.88 (m, 1H), 7.64 (m, 1H), 7.49 (m, 2H), 7.43 (m, 1H), 7.32 (m, 3H), 7.26 (m, 4H), 7.18 (m, 4H), 7.11 (m, 1H), 6.80 (s, 1H), 6.71 (m, 1H) ppm. ^13^C NMR (100 MHz, DMSO-*d*_6_): δ = 191.3, 176.5, 165.4, 156.0, 149.7, 149.4, 136.4, 135.1, 133.9, 129.9, 129.3 (2C), 128.4, 128.3, 128.1, 127.7, 126.3, 126.1, 125.3, 124.7 (2C), 122.8, 121.8, 116.6, 114.3, 78.7, 65.5, 62.8 ppm. IR (mineral oil): 1770, 1732, 1716, 1674 cm^−1^. Anal. Calcd (%) for 2C_29_H_18_N_2_O_5_S·C_7_H_8_: C 70.64; H 4.01; N 5.07. Found: C 70.82; H 4.11; N 5.00.*(3R*,3aS*,11aR*)-2,3-Diphenyl-3a-(thiophene-2-carbonyl)-3,3a-dihydrobenzo[d]pyrrolo[3′,4′:2,3]pyrrolo[2,1-b]thiazole-1,4,5(2H)-trione* (**3f**). The resulting precipitate was filtered off, washed with benzene (1 mL), and recrystallized from toluene (2–3 mL) to produce compound **3f**. Yield: 70 mg (45%); yellow solid; mp 169–171 °C. ^1^H NMR (400 MHz, DMSO-*d*_6_): δ = 8.15 (m, 1H), 7.87 (m, 1H), 7.51 (m, 3H), 7.40 (m, 1H), 7.35 (m, 1H), 7.31 (m, 4H), 7.26 (m, 2H), 7.23 (m, 2H), 7.21 (m, 2H), 7.17 (m, 1H), 7.12 (m, 1H), 6.75 (s, 1H) ppm. ^13^C NMR (100 MHz, DMSO-*d*_6_): δ = 191.2, 182.1, 165.0, 155.2, 140.7, 137.9, 136.2, 134.7, 133.8, 130.3, 129.2, 128.9, 128.5, 128.4 (2C), 128.1 (2C), 126.3, 126.1, 125.2, 124.4 (2C), 123.0, 117.1, 79.1, 65.9, 64.6 ppm. IR (mineral oil): 1762, 1715, 1650 cm^−1^. Anal. Calcd (%) for 2C_29_H_18_N_2_O_4_S_2_·C_7_H_8_: C 68.64; H 3.90; N 4.93. Found: C 68.83; H 4.11; N 4.99.*(3R*,3aS*,11aR*)-3a-Benzoyl-2-(3-nitrophenyl)-3-((E)-styryl)-3,3a-dihydrobenzo[d]pyrrolo[3′,4′:2,3]pyrrolo[2,1-b]thiazole-1,4,5(2H)-trione* (**3g**). The solvent was evaporated to 1 mL. The resulting precipitate was filtered off, washed with benzene (0.5 mL), and recrystallized from acetonitrile (1 mL) to produce compound **3g**. Yield: 20 mg (12%); yellow solid; mp 212–214 °C. ^1^H NMR (400 MHz, DMSO-*d*_6_): δ = 8.51 (m, 1H), 8.07 (m, 1H), 7.96 (m, 1H), 7.87 (m, 1H), 7.69 (m, 1H), 7.62 (m, 1H), 7.50 (m, 5H), 7.36 (m, 2H), 7.29 (m, 2H), 7.24 (m, 3H), 6.73 (d, J 15.7 Hz, 1H), 6.18 (d, J 9.3 Hz, 1H), 6.02 (dd, J 15.7 Hz, J 9.8 Hz, 1H) ppm. ^13^C NMR (100 MHz, DMSO-*d*_6_): δ = 193.3, 191.7, 165.4, 156.1, 147.9, 137.5, 137.4, 136.5, 136.0, 135.3, 134.1, 132.5, 130.4, 129.6 (2C), 129.2 (2C), 129.0, 128.6 (2C), 127.1 (2C), 126.9, 124.2, 123.6, 122.1, 121.6, 117.1, 78.9, 66.8, 64.2 ppm. IR (mineral oil): 1756, 1721, 1677 cm^−1^. Anal. Calcd (%) for C_33_H_21_N_3_O_6_S: C 67.45; H 3.60; N 7.15. Found: C 67.62; H 3.71; N 7.10.*(3R*,3aS*,11aR*)-3a-Benzoyl-3-(4-iodophenyl)-2-phenyl-3,3a-dihydrobenzo[d]pyrrolo[3′,4′:2,3]pyrrolo[2,1-b]thiazole-1,4,5(2H)-trione* (**3h**). The solvent was evaporated to 2.5 mL. The resulting precipitate was filtered off, washed with benzene (1 mL), and recrystallized from toluene (2–3 mL) to produce compound **3h**. Yield: 54 mg (28%); yellow solid; mp 170–172 °C. ^1^H NMR (400 MHz, DMSO-*d*_6_): δ = 7.85 (m, 1H), 7.59 (m, 3H), 7.51 (m, 3H), 7.47 (m, 4H), 7.36 (m, 1H), 7.30 (m, 3H), 7.14 (m, 3H), 6.68 (s, 1H) ppm. ^13^C NMR (100 MHz, DMSO-*d*_6_): δ = 192.4, 191.5, 165.0, 155.3, 137.0 (2C), 136.0, 135.6, 134.7, 133.8, 133.5, 131.0, 130.0, 128.9 (2C), 128.5 (2C), 128.2 (2C), 128.2, 127.8, 126.5, 126.2, 124.4 (2C), 123.0, 117.1, 95.2, 79.1, 66.3, 64.7 ppm. IR (mineral oil): 1762, 1715, 1691 cm^−1^. Anal. Calcd (%) for C_31_H_19_IN_2_O_4_S: C 57.95; H 2.98; N 4.36. Found: C 58.16; H 3.10; N 4.28.*(3R*,3aS*,11aR*)-3a-Benzoyl-3-(4-bromophenyl)-2-(4-methoxyphenyl)-3,3a-dihydrobenzo[d]pyrrolo[3′,4′:2,3]pyrrolo[2,1-b]thiazole-1,4,5(2H)-trione* (**3i**). The reaction mixture was evaporated to dryness. Then, the residue was recrystallized from benzene (2–3 mL) to produce compound **3i**. Yield: 52 mg (28%); yellow solid; mp 221–223 °C. ^1^H NMR (400 MHz, DMSO-*d*_6_): δ = 7.84 (m, 1H), 7.61 (m, 1H), 7.52 (m, 1H), 7.46 (m, 4H), 7.42 (m, 4H), 7.33 (m, 2H), 7.25 (m, 2H), 6.84 (m, 2H), 6.61 (s, 1H), 3.69 (s, 3H) ppm. ^13^C NMR (100 MHz, DMSO-*d*_6_): δ = 192.5, 191.7, 165.0, 157.5, 155.4, 135.7, 134.8, 133.7 (2C), 133.5, 131.3, 131.2 (2C), 130.1, 129.0 (2C), 128.8, 128.3 (2C), 126.3, 126.2 (2C), 123.1, 122.0, 117.1, 113.8 (2C), 79.1, 66.5, 64.9, 55.1 ppm. IR (mineral oil): 1761, 1729, 1710, 1689 cm^−1^. Anal. Calcd (%) for C_32_H_21_BrN_2_O_5_S: C 61.45; H 3.38; N 4.48. Found: C 61.63; H 3.50; N 4.40.*(3R*,3aS*,11aR*)-3a-Benzoyl-2-(4-chlorophenyl)-3-(3,4-dimethoxyphenyl)-3,3a-dihydrobenzo[d]pyrrolo[3′,4′:2,3]pyrrolo[2,1-b]thiazole-1,4,5(2H)-trione* (**3j**) *and (3S*,3aS*,11aR*)-3a-benzoyl-2-(4-chlorophenyl)-3-(3,4-dimethoxyphenyl)-3,3a-dihydrobenzo[d]pyrrolo[3′,4′:2,3]pyrrolo[2,1-b]thiazole-1,4,5(2H)-trione* (**3′j**). The reaction mixture was cooled to 10 °C. The resulting precipitate was filtered off (a mixture of products **3j** and **3′j**, 1:1). The mother liquor was evaporated to dryness. Then, the residue was recrystallized from ethanol (2–3 mL) at 60–65 °C to afform product **3j**. Product **3j**: Yield: 50 mg (28%); yellow solid; mp 238–240 °C. ^1^H NMR (400 MHz, DMSO-*d*_6_): δ = 7.84 (m, 1H), 7.62 (m, 1H), 7.58 (m, 2H), 7.50 (m, 5H), 7.34 (m, 4H), 6.80 (m, 3H), 6.56 (s, 1H), 3.69 (s, 3H), 3.60 (s, 3H) ppm. ^13^C NMR (100 MHz, DMSO-*d*_6_): δ = 193.0, 190.7, 165.0, 155.8, 148.9, 148.4, 135.6, 135.2, 134.8, 133.4, 130.6, 130.2, 128.7 (2C), 128.4 (2C), 128.4 (2C), 128.1, 126.2 (2C), 126.1, 125.5, 123.0, 121.4, 117.2, 112.5, 111.3, 79.4, 66.0, 65.2, 55.5, 55.2 ppm. IR (mineral oil): 1761, 1726, 1713, 1682 cm^−1^. Anal. Calcd (%) for C_33_H_23_ClN_2_O_6_S: C 64.86; H 3.79; N 4.58. Found: C 64.93; H 3.85; N 4.60. Product **3′j**: Yield: 24 mg (13%, the mixture **3j**:**3′j**, 1:1), orange solid; mp 224–226 °C (mixture **3j**:**3′j**, 1:1). ^1^H NMR (400 MHz, DMSO-*d*_6_): δ = 7.80 (m, 1H), 7.62 (m, 2H), 7.56 (m, 2H), 7.36 (m, 5H), 7.30 (m, 2H), 6.77 (m, 1H), 6.71 (m, 3H), 6.47 (s, 1H), 3.59 (s, 3H), 3.54 (s, 3H) ppm. ^13^C NMR (100 MHz, DMSO-*d*_6_): δ = 191.0, 168.1, 165.7, 154.4, 148.7, 148.1, 135.7, 134.9, 134.4, 132.7, 130.7, 130.1, 129.0 (2C), 128.6 (2C), 127.7 (2C), 127.6, 125.9, 125.7 (2C), 122.43, 121.0, 115.8, 112.8, 111.4, 79.2, 70.6, 65.2, 55.3, 55.2 ppm. IR (mineral oil): 1761, 1713, 1680 cm^−1^. Anal. Calcd. (%) for C_33_H_23_ClN_2_O_6_S: C 64.86; H 3.79; N 4.58. Found: C 64.12; H 3.94; N 4.71.*(3R*,3aS*,11aR*)-3a-Benzoyl-3-(3,4-dimethoxyphenyl)-2-phenyl-3,3a-dihydrobenzo[d]pyrrolo[3′,4′:2,3]pyrrolo[2,1-b]thiazole-1,4,5(2H)-trione* (**3k**) *and (3S*,3aS*,11aR*)-3a-benzoyl-3-(3,4-dimethoxyphenyl)-2-phenyl-3,3a-dihydrobenzo[d]pyrrolo[3′,4′:2,3]pyrrolo[2,1-b]thiazole-1,4,5(2H)-trione* (**3′k**). The reaction mixture was cooled to 10 °C. The resulting precipitate was filtered off and recrystallized from benzene (2–3 mL) to give product **3’k**. The mother liquor was evaporated to dryness, and the residue was recrystallized from toluene (2–3 mL) to produce product **3k**. Product **3k**: Yield: 40 mg (23%); yellow solid; mp 217–219 °C. ^1^H NMR (400 MHz, DMSO-*d*_6_): δ = 7.84 (m, 1H), 7.64 (m, 1H), 7.54 (m, 7H), 7.33 (m, 4H), 7.16 (m, 1H), 6.86 (s, 1H), 6.81 (m, 2H), 6.54 (s, 1H), 3.68 (s, 3H), 3.61 (s, 3H) ppm. ^13^C NMR (100 MHz, DMSO-*d*_6_): δ = 193.1, 190.6, 165.0, 155.9, 148.9, 148.5, 136.3, 135.4, 134.9, 133.5, 130.3, 128.7 (2C), 128.6 (2C), 128.5 (2C), 128.1, 126.4, 126.1, 125.8, 124.3 (2C), 123.0, 121.1, 117.3, 112.2, 111.3, 79.6, 65.7, 65.5, 55.5, 55.2 ppm. IR (mineral oil): 1759, 1728, 1715, 1689 cm^−1^. Anal. Calcd (%) for C_33_H_24_N_2_O_6_S: C 68.74; H 4.20; N 4.86. Found: C 68.90; H 4.27; N 4.79. Product **3′k**: Yield: 48 mg (28%); orange solid; mp 247–249 °C. ^1^H NMR (400 MHz, DMSO-*d*_6_): δ = 7.82 (m, 3H), 7.62 (m, 1H), 7.52 (m, 3H), 7.40 (m, 2H), 7.32 (m, 4H), 7.16 (m, 1H), 6.73 (m, 2H), 6.67 (m, 1H), 6.46 (s, 1H), 3.59 (s, 3H), 3.52 (s, 3H) ppm. ^13^C NMR (100 MHz, DMSO-*d*_6_): δ = 191.4, 188.0, 165.8, 154.5, 148.7, 148.0, 136.0, 135.8, 134.5, 132.9, 130.3, 129.2 (2C), 128.7 (2C), 127.9 (2C), 126.6, 126.1, 125.8, 124.1 (2C), 123.0, 122.5, 121.0, 115.9, 112.7, 111.2, 79.3, 70.8, 64.5, 55.3, 55.2 ppm. IR (mineral oil): 1762, 1709, 1669 cm^−1^. Anal. Calcd (%) for C_33_H_24_N_2_O_6_S: C 68.74; H 4.20; N 4.86. Found: C 68.86; H 4.31; N 4.93.*(3R*,3aS*,11aR*)-3a-Benzoyl-2-benzyl-3-(4-bromophenyl)-3,3a-dihydrobenzo[d]pyrrolo[3′,4′:2,3]pyrrolo[2,1-b]thiazole-1,4,5(2H)-trione* (**3l**). The solvent was evaporated. The resulting mass was stirred in acetonitrile (2 mL) at 80 °C for 5 min. Then, the obtained mixture was cooled to room temperature, and the formed precipitate was filtered off to produce compound **3l**. Yield: 60 mg (33%); yellow solid; mp 204–206 °C. ^1^H NMR (400 MHz, DMSO-*d*_6_): δ = 7.79 (m, 1H), 7.59 (m, 3H), 7.54 (m, 1H), 7.37 (m, 9H), 7.16 (m, 2H), 7.09 (m, 2H), 5.58 (s, 1H), 4.95 (d, J 15.2 Hz, 1H), 3.71 (d, J 14.7 Hz, 1H) ppm. ^13^C NMR (100 MHz, DMSO-*d*_6_): δ = 192.7, 191.2, 166.1, 155.1, 134.7, 134.6, 134.0, 133.8, 132.6, 131.9 (2C), 130.7, 129.8, 128.8 (2C), 128.7 (2C), 128.3 (2C), 128.3, 128.1 (2C), 128.0, 127.6, 126.3, 123.2, 122.6, 117.3, 78.9, 65.6, 63.9, 45.4 ppm. IR (mineral oil): 1760, 1719, 1672 cm^−1^. Anal. Calcd (%) for C_32_H_21_BrN_2_O_4_S: C 63.06; H 3.47; N 4.60. Found: C 63.14; H 3.58; N 4.63.*(3R*,3aS*,11aR*)-3a-Benzoyl-2-(benzylideneamino)-3-phenyl-3,3a-dihydrobenzo[d]pyrrolo[3′,4′:2,3]pyrrolo[2,1-b]thiazole-1,4,5(2H)-trione* (**3m**). The solvent was evaporated to 1 mL. The resulting precipitate was filtered off, stirred in benzene (2–3 mL) at 85 °C for 10 min, and then filtered off to produce compound **3m**. Yield: 42 mg (26%); yellow solid; mp 224–226 °C. ^1^H NMR (400 MHz, DMSO-*d*_6_): δ = 8.63 (s, 1H), 7.81 (m, 1H), 7.65 (m, 1H), 7.55 (m, 7H), 7.43 (m, 3H), 7.35 (m, 5H), 7.29 (2H), 6.54 (s, 1H) ppm. ^13^C NMR (100 MHz, DMSO-*d*_6_): δ = 192.4, 190.4, 162.3, 156.6, 155.5, 134.9, 134.6, 133.7, 133.5, 132.9, 131.3, 130.4, 129.0, 128.9 (2C), 128.8 (2C), 128.8 (2C), 128.7 (2C), 128.3, 128.0 (2C), 127.5 (2C), 126.1, 123.0, 117.5, 78.3, 65.3, 64.7 ppm. IR (mineral oil): 1757, 1725, 1686 cm^−1^. Anal. Calcd (%) for C_32_H_21_N_3_O_4_S: C 70.71; H 3.89; N 7.73. Found: C 70.93; H 3.97; N 7.81.

#### 3.1.3. Procedure to Compounds **8a**–**j**

A mixture of APBTT **1** (0.298 mmol) and carbodiimide **7** (0.313 mmol) in toluene (5 mL) was heated for 1 h at 115 °C (until the dark violet color characteristic of APBTT **1** disappeared and a transparent yellow solution formed). Then, the reaction mixture was cooled to room temperature. The resulting precipitate was filtered off and stirred in ethanol (2 mL) at 45–50 °C for 30 min. Then, the precipitate was filtered off and recrystallized from toluene (5 mL) to produce the corresponding compound **8**.

*(3aS*,11aR*)-3a-Benzoyl-2-cyclohexyl-3-(cyclohexylimino)-3,3a-dihydrobenzo[d]pyrrolo[3′,4′:2,3]pyrrolo[2,1-b]thiazole-1,4,5(2H)*-trione (**8a**). Yield: 69 mg (44%); yellow solid; mp 264–266 °C. ^1^H NMR (400 MHz, CDCl_3_): δ = 7.87 (m, 1H), 7.68 (m, 3H), 7.49 (m, 2H), 7.25 (m, 3H), 4.31 (m, 1H), 3.41 (m, 1H), 2.32 (m, 2H), 1.86 (s, 3H), 1.73 (m, 4H), 1.51 (m, 1H), 1.42 (m, 3H), 1.36 (m, 2H), 1.21 (m, 2H), 0.91 (m, 2H), 0.38 (m, 1H) ppm. ^13^C NMR (100 MHz, CDCl_3_): δ = 190.7, 187.7, 168.0, 154.1, 141.6, 135.1, 134.8, 133.8, 129.3, 129.1 (2C), 128.7 (2C), 128.5, 126.9, 122.6, 118.0, 77.6, 65.4, 60.6, 54.0, 34.1, 32.1, 28.2, 27.8, 25.9, 25.8, 25.7, 25.2, 23.9, 23.8 ppm. IR (mineral oil): 1779, 1749, 1722, 1677 cm^−1^. Anal. Calcd (%) for C_31_H_31_N_3_O_4_S: C 68.74; H 5.77; N 7.76. Found: C 68.85; H 5.83; N 7.71.*(3aS*,11aR*)-2-Cyclohexyl-3-(cyclohexylimino)-3a-(4-methylbenzoyl)-3,3a-dihydrobenzo[d]pyrrolo[3′,4′:2,3]pyrrolo[2,1-b]thiazole-1,4,5(2H)-trione* (**8b**). Yield: 25 mg (15%); yellow solid; mp 230–232 °C. ^1^H NMR (400 MHz, CDCl_3_): δ = 7.87 (m, 1H), 7.59 (m, 2H), 7.29 (m, 2H), 7.24 (m, 3H), 4.33 (m, 1H), 3.45 (m, 1H), 2.46 (s, 3H), 2.31 (m, 2H), 1.86 (m, 3H), 1.75 (m, 1H), 1.69 (m, 4H), 1.53 (m, 1H), 1.44 (m, 3H), 1.39 (m, 1H), 1.35 (m, 1H), 1.29 (m, 1H), 1.20 (m, 2H), 0.93 (m, 2H), 0.47 (m, 1H) ppm. ^13^C NMR (100 MHz, CDCl_3_): δ = 190.1, 187.8, 168.1, 154.1, 146.6, 141.8, 134.8, 131.2, 129.8, 129.4, 129.0, 128.9, 128.5, 128.2, 126.8, 125.3, 122.5, 118.0, 77.7, 60.5, 54.0, 34.1, 32.2, 28.2, 27.8, 25.9, 25.8, 25.7, 25.2, 24.0, 23.7, 21.8 ppm. IR (mineral oil): 1788, 1744, 1729, 1674 cm^−1^. Anal. Calcd (%) for C_32_H_33_N_3_O_4_S: C 69.17; H 5.99; N 7.56. Found: C 69.55; H 6.12; N 7.66.*(3aS*,11aR*)-2-Cyclohexyl-3-(cyclohexylimino)-3a-(4-fluorobenzoyl)-3,3a-dihydrobenzo[d]pyrrolo[3′,4′:2,3]pyrrolo[2,1-b]thiazole-1,4,5(2H)-trione* (**8c**). Yield: 18 mg (11%); yellow solid; mp 228–230 °C. ^1^H NMR (400 MHz, CDCl_3_): δ = 7.92 (m, 1H), 7.79 (m, 2H), 7.33 (m, 2H), 7.23 (m, 3H), 4.36 (m, 1H), 3.47 (m, 1H), 2.36 (m, 2H), 1.92 (m, 3H), 1.80 (m, 1H), 1.74 (m, 3H), 1.60–1.47 (m, 4H), 1.45–1.37 (m, 2H), 1.29–1.19 (m, 2H), 1.00 (m, 2H), 0.52 (m, 1H) ppm. ^13^C NMR (100 MHz, CDCl_3_): δ = 189.1, 187.4, 168.0, 166.7 (d, *J* = 261.6 Hz), 154.0, 141.5, 134.7, 131.5 (d, *J* = 10.1 Hz, 2C), 130.2 (d, *J* = 3.0 Hz), 129.1, 128.6, 127.0, 122.6, 118.1, 116.5 (d, *J* = 24.2 Hz, 2C), 77.5, 65.3, 60.6, 54.1, 34.1, 32.3, 28.3, 27.8, 25.9, 25.8, 25.6, 25.1, 23.9, 23.8 ppm. ^19^F NMR (376 MHz, DMSO-*d*_6_): δ = −100.03 ppm. IR (mineral oil): 1789, 1749, 1726, 1673 cm^−1^. Anal. Calcd (%) for C_31_H_30_FN_3_O_4_S: C 66.53; H 5.40; N 7.51. Found: C 66.68; H 5.51; N 7.60.*(3aS*,11aR*)-3a-(4-Bromobenzoyl)-2-cyclohexyl-3-(cyclohexylimino)-3,3a-dihydrobenzo[d]pyrrolo[3′,4′:2,3]pyrrolo[2,1-b]thiazole-1,4,5(2H)-trione* (**8d**). Yield: 57 mg (31%); yellow solid; mp 180–182 °C. ^1^H NMR (400 MHz, CDCl_3_): δ = 7.92 (m, 1H), 7.71 (m, 2H), 7.61 (m, 2H), 7.32 (m, 1H), 7.26 (m, 2H), 4.35 (m, 1H), 3.46 (m, 1H), 2.36 (m, 2H), 1.91 (m, 3H), 1.80 (m, 1H), 1.74 (m, 2H), 1.60 (m, 3H), 1.48 (m, 1H), 1.46–1.31 (m, 3H), 1.25 (m, 2H), 1.02 (m, 2H), 0.55 (m, 1H) ppm. ^13^C NMR (100 MHz, CDCl_3_): δ = 189.6, 187.3, 167.9, 153.9, 141.3, 134.7, 132.5 (2C), 130.7, 130.0 (2C), 129.0, 128.6, 127.0, 122.6, 118.1, 77.4, 65.2, 60.5, 54.1, 34.1, 32.3, 28.2, 27.8, 25.8, 25.8, 25.6, 25.1, 23.9, 23.8 ppm. IR (mineral oil): 1791, 1789, 1752, 1726, 1672 cm^−1^. Anal. Calcd (%) for C_31_H_30_BrN_3_O_4_S: C 60.00; H 4.87; N 6.77. Found: C 60.14; H 4.97; N 6.65.*(3aS*,11aR*)-2-Cyclohexyl-3-(cyclohexylimino)-3a-(furan-2-carbonyl)-3,3a-dihydrobenzo[d]pyrrolo[3′,4′:2,3]pyrrolo[2,1-b]thiazole-1,4,5(2H)-trione* (**8e**). Yield: 71 mg (45%); yellow solid; mp 251–253 °C. ^1^H NMR (400 MHz, CDCl_3_): δ = 7.91 (m, 1H), 7.60 (s, 1H), 7.51 (m, 1H), 7.28 (m, 3H), 6.72 (s, 1H), 4.34 (m, 1H), 3.60 (br.s, 1H), 2.34 (m, 2H), 1.85 (m, 4H), 1.73 (d, J = 12, 3H), 1.63 (m, 2H), 1.52 (m, 2H), 1.38 (m, 3H), 1.22 (m, 3H), 0.95–0.62 (m, 1H) ppm. ^13^C NMR (100 MHz, CDCl_3_): δ = 186.6, 168.2, 149.4 (2C), 140.5, 134.9, 129.3, 128.4, 126.7, 122.6, 120.9, 118.0, 114.2, 77.7, 77.2, 54.1, 34.0, 28.3, 27.8, 25.9, 25.9, 25.7, 25.2, 24.0, 23.8 ppm. IR (mineral oil): 1775, 1745, 1721, 1667 cm^−1^. Anal. Calcd (%) for C_29_H_29_N_3_O_5_S: C 65.52; H 5.50; N 7.90. Found: C 65.71; H 5.58; N 8.03.*(3aS*,11aR*)-2-Cyclohexyl-3-(cyclohexylimino)-3a-(thiophene-2-carbonyl)-3,3a-dihydrobenzo[d]pyrrolo[3′,4′:2,3]pyrrolo[2,1-b]thiazole-1,4,5(2H)-trione* (**8f**). Yield: 96 mg (59%); yellow solid; mp 274–276 °C. ^1^H NMR (400 MHz, CDCl_3_): δ = 7.91 (m, 2H), 7.55 (m, 1H), 7.32 (m, 2H), 7.23 (m, 2H), 4.39 (m, 1H), 3.70 (m, 1H), 2.38 (m, 2H), 1.94 (m, 3H), 1.83 (m, 1H), 1.76 (m, 3H), 1.60 (m, 2H), 1.54 (m, 2H), 1.42 (m, 2H), 1.29 (m, 2H), 1.13 (m, 2H), 0.75 (m, 1H) ppm. ^13^C NMR (100 MHz, CDCl_3_): δ = 170.7, 168.0, 154.0, 141.3, 137.4 (2C), 134.8, 133.1, 129.2, 128.6, 128.5, 126.9, 122.6, 118.0, 77.8, 77.2, 60.4, 54.1, 34.2, 32.3, 28.3, 27.8, 25.9, 25.8, 25.7, 25.2, 24.0, 23.8 ppm. IR (mineral oil): 1779, 1747, 1724, 1682, 1657 cm^−1^. Anal. Calcd (%) for C_29_H_29_N_3_O_4_S_2_: C 63.60; H 5.34; N 7.67. Found: C 63.83; H 5.41; N 7.54.*(3aS*,11aR*)-3a-(4-Chlorobenzoyl)-2-cyclohexyl-3-(cyclohexylimino)-3,3a-dihydrobenzo[d]pyrrolo[3′,4′:2,3]pyrrolo[2,1-b]thiazole-1,4,5(2H)-trione* (**8g**). Yield: 43 mg (25%); yellow solid; mp 248–250 °C. ^1^H NMR (400 MHz, CDCl_3_): δ = 7.92 (m, 1H), 7.69 (m, 2H), 7.53 (m, 2H), 7.30 (m, 3H), 4.35 (m, 1H), 3.46 (m, 1H), 2.36 (m, 2H), 1.92 (m, 3H), 1.80 (m, 1H), 1.74 (m, 3H), 1.58 (m, 2H), 1.48 (m, 2H), 1.39 (m, 2H), 1.26 (m, 2H), 1.01 (m, 2H), 0.53 (m, 1H) ppm. ^13^C NMR (100 MHz, CDCl_3_): δ = 189.4, 187.4, 167.9, 153.9, 142.0, 141.3, 134.7, 132.1, 130.0 (2C), 129.5 (2C), 129.1, 128.6, 127.0, 122.6, 118.1, 77.5, 65.3, 60.5, 54.1, 34.1, 32.3, 28.3, 27.8, 25.9, 25.8, 25.6, 25.1, 23.9, 23.8 ppm. IR (mineral oil): 1791, 1789, 1751, 1726, 1674 cm^−1^. Anal. Calcd (%) for C_31_H_30_ClN_3_O_4_S: C 64.63; H 5.25; N 7.29. Found: C 64.69; H 5.19; N 7.37.*(3aS*,11aR*)-3a-Benzoyl-2-isopropyl-3-(isopropylimino)-3,3a-dihydrobenzo[d]pyrrolo[3′,4′:2,3]pyrrolo[2,1-b]thiazole-1,4,5(2H)-trione* (**8h**). Yield: 89 mg (65%); yellow solid; mp 207–207 °C. ^1^H NMR (400 MHz, CDCl_3_): δ = 7.94 (m, 1H), 7.74 (m, 3H), 7.56 (m, 2H), 7.35–7.25 (m, 3H), 4.78 (m, 1H), 3.79 (m, 1H), 1.54 (m, 6H), 1.30 (m, 3H), 0.50 (d, J = 8 Hz, 3H) ppm. ^13^C NMR (100 MHz, CDCl_3_): δ = 190.3, 187.8, 167.9, 154.1, 141.4, 135.2, 134.8, 133.8, 129.2, 129.2 (2C), 128.8 (2C), 128.5, 126.9, 122.6, 118.1, 77.6, 66.7, 52.8, 46.1, 24.4, 22.0, 18.8, 18.4 ppm. IR (mineral oil): 1786, 1749, 1724, 1669 cm^−1^. Anal. Calcd (%) for C_25_H_23_N_3_O_4_S: C 65.06; H 5.02; N 9.10. Found: C 65.17; H 5.09; N 9.18.*(3aS*,11aR*)-2-Isopropyl-3-(isopropylimino)-3a-(4-methylbenzoyl)-3,3a-dihydrobenzo[d]pyrrolo[3′,4′:2,3]pyrrolo[2,1-b]thiazole-1,4,5(2H)-trione* (**8i**). Yield: 106 mg (75%); yellow solid; mp 216–218 °C. ^1^H NMR (400 MHz, CDCl_3_): δ = 7.94 (m, 1H), 7.67 (m, 2H), 7.37 (m, 2H), 7.35–7.25 (m, 3H), 4.79 (m, 1H), 3.84 (m, 1H), 2.52 (s, 3H), 1.55 (m, 6H), 1.31 (m, 3H), 0.53 (m, 3H) ppm. ^13^C NMR (100 MHz, CDCl_3_): δ = 189.8, 187.8, 167.9, 154.1, 146.8, 141.6, 134.8, 131.2, 129.9, 129.3, 129.0, 128.5, 126.9 (2C), 122.6, 118.0 (2C), 115.0, 52.7, 46.1, 24.4, 22.1, 21.8, 18.8, 18.4 ppm. IR (mineral oil): 1786, 1751, 1727 cm^−1^. Anal. Calcd (%) for C_26_H_25_N_3_O_4_S: C 65.67; H 5.30; N 8.84. Found: C 65.81; H 5.26; N 8.80.*(3aS*,11aR*)-3a-(4-Chlorobenzoyl)-2-isopropyl-3-(isopropylimino)-3,3a-dihydrobenzo[d]pyrrolo[3′,4′:2,3]pyrrolo[2,1-b]thiazole-1,4,5(2H)-trione* (**8j**). Yield: 106 mg (72%); yellow solid; mp 228–230 °C. ^1^H NMR (400 MHz, CDCl_3_): δ = 7.64 (m, 1H), 7.41 (m, 2H), 7.30 (m, 1H), 7.06–6.96 (m, 4H), 4.48 (m, 1H), 3.49 (m, 1H), 1.25 (d, J = 4 Hz, 6H), 1.02 (d, J = 8 Hz, 3H), 0.29 (d, J = 8 Hz, 3H) ppm. ^13^C NMR (100 MHz, CDCl_3_): δ = 189.1, 187.4, 167.7, 153.9, 142.1, 141.2, 134.7, 132.0, 131.3, 130.1 (2C), 129.6, 129.0, 128.6 (2C), 127.0, 122.6, 118.1, 52.8, 46.2, 24.4, 22.2, 18.8, 18.4 ppm. IR (mineral oil): 1786, 1750, 1726, 1664 cm^−1^. Anal. Calcd (%) for C_25_H_22_ClN_3_O_4_S: C 60.54; H 4.47; N 8.47. Found: C 60.76; H 4.58; N 8.55.

#### 3.1.4. Procedures to Compounds **4a**,**b**,**6d**

*5-Hydroxy-4-(2-oxo-2H-benzo[b][1,4]thiazin-3(4H)-ylidene)-1,5-diphenylpyrrolidine-2,3-dione* (**4a**). Method 1. Acetic acid (3 mL) was added to the mixture of APBTT **1a** (100 mg, 0.298 mmol) and *N*-benzylideneaniline **2a** (54 mg, 0.298 mmol). The mixture was stirred for 24 h at room temperature. The resulting precipitate was filtered off and washed with acetone (2 mL) to produce compound **4a**. Yield: 110 mg (86%); red solid; mp 179–181 °C. Method 2. Aniline **5a** (27.2 μL, 0.298 mmol) was added to the mixture of acetic acid (3 mL) and APBTT **1a** (100 mg, 0.298 mmol). The mixture was stirred for 24 h at room temperature. The resulting precipitate was filtered off and washed with acetone (2 mL) to produce compound **4a**. Yield: 121 mg (95%); red solid; mp 179–181 °C. ^1^H NMR (400 MHz, DMSO-*d_6_*): δ = 14.05 (s, 1H), 7.60 (m, 1H), 7.52 (m, 2H), 7.38 (m, 1H), 7.34 (m, 2H), 7.22 (m, 3H), 7.18 (m, 1H), 7.16 (s, 1H), 7.13 (m, 1H), 7.11 (s, 1H), 7.04 (m, 2H) ppm. IR (mineral oil): 3467, 3169, 3083, 1723, 1684, 1632 cm^−1^. Anal. Calcd (%) for C_24_H_16_N_2_O_4_S: C 67.28; H 3.76; N 6.54. Found: C 67.39; H 3.81; N 6.47.*1-Benzyl-5-hydroxy-4-(2-oxo-2H-benzo[b][1,4]thiazin-3(4H)-ylidene)-5-(4-methylphenyl)pyrrolidine-2,3-dione* (**4b**). Benzylamine **5b** (15.6 μL, 0.143 mmol) was added to the mixture of acetic acid (3 mL) and APBTT **1a** (50 mg, 0.143 mmol). The mixture was stirred for 24 h at room temperature. The resulting precipitate was filtered off and washed with acetone (1 mL) to produce compound **4b**. Yield: 63 mg (46%); orange solid; mp 162–164 °C. ^1^H NMR (400 MHz, DMSO-*d_6_*): δ = 14.24 (s, 1H), 7.55 (m, 1H), 7.46 (m, 2H), 7.31 (m, 3H), 7.12 (m, 3H), 7.03 (m, 4H), 6.67 (s, 1H), 4.20 (dd, *J* 64.1 Hz, *J* 15.2 Hz, 2H), 2.24 (s, 3H) ppm. ^13^C NMR (400 MHz, DMSO-*d_6_*): δ = 183.1, 177.4, 160.4, 138.8, 137.8, 137.0, 136.3, 129.5, 128.6, 128.1 (2C), 127.8 (2C), 127.5 (2C), 126.3, 126.1, 125.9 (2C), 125.5, 120.5, 119.4, 109.9, 88.6, 42.5, 20.5 ppm. IR (mineral oil): 3474, 3192, 3058, 1720, 1640 cm^−1^. Anal. Calcd (%) for C_26_H_20_N_2_O_4_S: C 68.41; H 4.42; N 6.14. Found: C 68.56; H 4.47; N 6.20.*(Z)-3-(Benzo[d]thiazol-2(3H)-ylidene)-4-(4-bromophenyl)-2,4-dioxo-N-phenylbutanamide* (**6d**). This product was a by-product in the synthesis of compound **3d**. Product **6d** was precipitated in the first fraction upon recrystallization of the main product from toluene. Yield: 3 mg (2%); colorless crystals; mp 179–181 °C. ^1^H NMR (400 MHz, CDCl_3_): δ = 13.76 (br.s, 1H), 10.39 (s, 1H), 8.07 (m, 1H), 7.53 (m, 3H), 7.43 (m, 3H), 7.29 (m, 2H), 7.21 (m, 3H), 7.02 (m, 1H) ppm. IR (mineral oil): 3278, 3136, 1718, 1672, 1635 cm^−1^. Anal. Calcd (%) for C_23_H_15_BrN_2_O_3_S: C 57.63; H 3.15; N 5.84. Found: C 57.86; H 3.24; N 5.91.

### 3.2. Computational Details

The DFT calculations for all model structures were carried out at the M06-2X/6-31G* level of theory with the help of the Gaussian-09 program package [81]. No symmetry restrictions have been applied during the geometry optimization procedure. The Hessian matrices were calculated analytically for all optimized model structures to prove the location of correct minima or saddle points (transition states) on the potential energy surface. The Cartesian atomic coordinates for all model structures are presented in attached xyz-files (Appendix A).

### 3.3. Biology

#### 3.3.1. Screening of Substances in 96-Well Plates

*Chlorella vulgaris* (strain IMBR-19, obtained from the A.O. Kovalevsky Institute of Biology of the Southern Seas of RAS, Sevastopol, Russia) was cultivated in BG-11 medium. BG-11 medium was prepared using following solutions: Solution #1: Na_2_-EDTA—20 mg in 20 mL; Solution #2: citric acid—120 mg and iron(III) citrate—120 mg in 20 mL; Solution #3: K_2_HPO_4_∙3H_2_O—800 mg in 20 mL; Solution #4. MgSO_4_∙7H_2_O—1.5 g in 20 mL; Solution #5: CaCl_2_∙2H_2_O—720 mg in 20 mL; Solution #6: Na_2_CO_3_—400 mg in 20 mL; Solution #7: NaNO_3_—15 g in 100mL; Solution #8: H_3_BO_3_—57.2 mg, MnCl_2_∙4H_2_O—36.2 mg, ZnSO_4_∙7H_2_O—4.4 mg, CuSO_4_∙5H_2_O—1.58 mg, Na_2_MoO_4_∙2H_2_O—7.8 mg, and 1 mL of 0.988 g/L Co(NO_3_)_2_∙6H_2_O in 20 mL. All of the solutions were prepared using Milli-Q water. To obtain 500 mL of BG-11 medium, 500 µL of solutions #1–6 and #8 and 5 mL of solution #7 were added to 400 mL of water. Then, the volume was adjusted to 500 mL. A solution of D(+)-glucose in BG-11 (6 g/L) was used for the preparation of positive controls.

Cultures of *C. vulgaris* were maintained and cultivated in aseptic conditions, with the only exception being DMSO solutions of tested substances that were not sterilized upon dilution.

Substances to be tested were diluted to 1 × 10^−3^, 1 × 10^−4^, and 1 × 10^−5^ mol/L in 99% DMSO before the experiment. Poorly soluble substances were either kept for 16 h on a rotator or treated by ultrasound using an ultrasound homogenizer equipped with a 3 mm probe (VCX-130, Sonics and Materials, Newtown, CT, USA).

Starter cultures of *C. vulgaris* were prepared as follows: stock culture cells (2 mL, in exponential growth phase) were washed two times with BG-11 medium by centrifugation (20 min, 350 g). The cells were then diluted in 10 mL of BG-11, and the cell concentration was measured using a hemocytometer.

In the wells of 96-well culture plates, BG-11 medium, starter culture of *C. vulgaris*, and tested substances diluted in DMSO were combined, resulting in a total volume of 300 µL. The resulting number of cells was 5 × 10^4^ cells/well, with a resulting volume fraction of DMSO at 1%. The resulting concentrations of tested substances were 1 × 10^−5^, 1 × 10^−6^, and 1 × 10^−7^ mol/L. In negative control and positive control wells, pure DMSO was added. BG-11 with glucose was added to the positive control wells, resulting in a concentration of glucose of 2 g/L. All substances were tested in duplicate (2 wells for each concentration). In the edge and corner wells of the plates, sterile distilled water was added. The plates were sealed with a gas-permeable film.

All cultures were maintained for 5 days in a humid chamber at +28 °C at 150 rpm under cyclic illumination consisting of 12 h on: 12 h off. The light intensity was 100 μmol·m^−2^·s^−1^. The lighting unit was an array of evenly distributed white LEDs with a cooling device preventing well heating, positioned below the culture plates (bottom illumination).

After the end of cultivation, the contents of the wells were mixed using a multichannel pipette, and the cell concentration was assessed by measuring the absorbance at 750 nm.

#### 3.3.2. Evaluation of Lead Substances in 50-mL Flasks

Substances to be tested were diluted to 1 × 10^−2^, 1 × 10^−3^, 1 × 10^−4^, and 1 × 10^−5^ mol/L in 99% DMSO before the experiment. Starter cultures of *C. vulgaris* were prepared as follows: stock culture cells (12 mL, in exponential growth phase) were washed twice with BG-11 medium by centrifugation (15 min, 450 g). The cells were then diluted in 10 mL of BG-11, and the cell concentration was measured using a hemocytometer.

In the 50-mL Erlenmeyer flasks, BG-11 medium, the starter culture of *C. vulgaris*, and tested substances diluted in DMSO were combined, resulting in a total volume of 30 mL. The resulting number of cells was 1 × 10^7^ cells/flask, with a resulting volume fraction of DMSO at 1%. The resulting concentrations of tested substances were 1 × 10^−4^, 1 × 10^−5^, 1 × 10^−6^, and 1 × 10^−7^ mol/L. In the negative control and positive control flasks, pure DMSO was added. BG-11 with glucose was added to the positive control flasks, resulting in a concentration of glucose of 2 g/L. All substances were tested in triplicate (three flasks for each concentration). The flasks were sealed with gas-permeable cellulose caps. All cultures were maintained for 5 days in a humid chamber at +28 °C at 150 rpm under cyclic illumination consisting of 12 h on: 12 h off. The light intensity was 100 μmol·m^−2^·s^−1^. The lighting unit was an array of evenly distributed white LEDs with a cooling device preventing overheating, positioned below the culture plates (bottom illumination).

#### 3.3.3. Cell Count and Pigments Analysis

The flask contents were carefully mixed, and then 10 mL of cell culture was transferred into 15-mL centrifuge tubes. The cells were washed twice with 10 mL of water by centrifugation (15 min, 450 g). The washed cells were diluted in 10 mL of water, and the cell concentration was measured using a hemocytometer.

Pigment extraction was carried out as follows: two milliliters of cell culture were transferred to centrifuge tubes. The cells were washed by centrifugation two times at 7000× *g* for 10 min and concentrated two-fold. After the second wash, the sediment was vortexed for 1 min, and then 90% methanol was added. The tubes were heated at +60 °C for 30 min in a solid-state thermostat. Then, the samples were cooled to room temperature and centrifuged at 10,000× *g* for 10 min. The absorbance of the supernatant containing extracted pigments was measured at 665, 652, and 470 nm. The concentrations of chlorophylls and carotenoids were calculated as described in the papers [82,83]. After that, the concentration of pigments in micrograms per 1 × 10^7^ cells was calculated.

## 4. Conclusions

An approach to a new 6/5/5/5-tetracyclic alkaloid-like spiroheterocyclic system of benzo[*d*]pyrrolo[3′,4′:2,3]pyrrolo[2,1-*b*]thiazole **3**, **8** was developed on the basis of a reaction of 3-aroylpyrrolo[2,1-*c*][1,4]benzothiazine-1,2,4-triones **1** with Schiff bases **2** and carbodiimides **7**. This reaction proceeded as a nucleophile-induced ring contraction—intramolecular cyclization cascade. The formation of the benzo[*d*]pyrrolo[3′,4′:2,3]pyrrolo[2,1-*b*]thiazoles **3**, **8** was found to be diastereoselective, with the exception of compounds **3k**,**j**, **3′k**,**j**. Compounds **3a**, **8j** were found to promote the growth of *Chlorella vulgaris* and to increase the chlorophyll content in its cells.

## 5. Patents

The method for preparing products **8** has been patented [84].

## Data Availability

The presented data are available in this article.

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
