# Peer review of "Reaction of Pyrrolobenzothiazines with Schiff Bases and Carbodiimides: Approach to Angular 6/5/5/5-Tetracyclic Spiroheterocycles"

_molecules, 2024, doi:10.3390/molecules29092089_

Round 1

Reviewer 1 Report

Comments and Suggestions for Authors

please rewrite the abstract in a more scientific way.

please rewrite the introduction section. it is needed to bring some examples from these types of compounds with biological or chemical properties and applications. introduction must be more comprehensive.

please explain the purpose of your research much more clearly.

Author Response

The abstract was written according to the MDPI recommendations. What issues exactly should be changed?

The introduction already gives examples of compounds with biological or chemical properties and applications in Fig. 2.

The purpose of our research was to develop a synthetic approach to an unprecedented 6/5/5/5-tetracyclic alkaloid-like spiroheterocyclic system of benzo[d]pyrrolo[3',4':2,3]pyrrolo[2,1-b]thiazole via the reaction of pyrrolobenzothiazines with Schiff bases and carbodiimides. It was clearly stated in the manuscript.

Reviewer 2 Report

Comments and Suggestions for Authors

This manuscript described a new method to synthesize a unique alkaloid-like compound, benzopyrrolo[3',4':2,3]pyrrolo[2,1-b]thiazole, by reacting pyrrolobenzothiazines with Schiff bases and carbodiimides. Experimental and computational analyses confirm the reaction mechanism and stereoselectivity. Additionally, these compounds show promise as growth stimulants and pigment promoters in Chlorella vulgaris algae.

1. Please explain the role of acetic acid in Scheme 5. 

Author Response

AcOH is the reaction solvent (protic) and possibly a catalyst which is discussed in the manuscript.

Reviewer 3 Report

Comments and Suggestions for Authors

1) Figure 1: names of alkaloids (or related) should be added into the figure

2) Table 1: in the last column, instead of a foot note, the title should be: HPLC yield

3) Line 106: is any other by-product identified? Did the authors check consumption of the Schiff base? The monitoring of the reaction was done visually by violet colour disappearance related to the APBTT, can this disappearance of colour be related only to ABPTT decomposition, instead of unselectively reaction with Schiff base?

4) Table 2: i) the entries should be correlatively numbered (as in Table 1): 1, 2, 3…13. The code of the final compounds should be indicated in the last column along with the yied; ii) the Schiff bases that were not successful should also be incorporated in the table (at least as footnote, see comment N° 7); iii) n/d abbreviation should be indicated as a table footnote

5) Lines 151-156: “For these reasons, we replaced chloroform with benzene in these reactions. As a result, we noticed a decreasing trend in the HPLC yields of products 3 (Conditions B, Table 2), which correlated with the optimization data for the test reaction of APBTT 1a with Schiff base 2a (Table 1). Moreover, products 3 were easily isolated and purified by simple crystallization directly from the reaction mixtures (in scale of 298 μmol, benzene).”

This analysis paragraph should be re-written: i) “…we noticed a decreasing tendency in the HPLC yields of products 3 for most of the substrates evaluated (compared conditions in entries xxx, table 2)…” ii) Moreover should be changed by “however”.

Did the authors try or evaluate the possibility of non-polar solvent with better properties regarding environment issues (such as cymene)

6) Regarding the issue discussed above: is the yield measured by HPLC (table 2) representative of the efficacy of the reaction? Could it be possible that part of the product is already precipitated in benzene and, therefore, less product is detected, which would mean that in both tested solvents the formation of the product is comparable, but the isolated yields (because of possible precipitation-crystallization) are better for benzene?

7) lines 159-165: the examples that were not successful in the studied transformation should be also added in Table 2 as this is also important information to analysis the scope of the reaction. In this paragraph, “APBTTs 1” should be replaced by “APBTTs 1a-g” (if all of them were evaluated, or to detail which of them were tested). Incorporation of these experiments in Table 2 will help to clarify this issue.

8) lines 181-182: “In our scope studies (Table 2), we isolated exclusively (3R*,3aS*,11aR*)-diastereomers for compounds 3a-i,l,m.” In the reactions leading to 3a-i,l,m only, were (3R*,3aS*,11aR*)-diastereomers isolated nor detected? In the HPLC analysis (previous to purification) was any other diastereoisomer detected? Were the crude reaction mixtures evaluated by a different method rather than HPLC? The main text should be clear about this issue: if the other diastereoisomers were not isolated, or,  if they were not detected nor identified in the reaction mixtures, if analyzed.

9) line 189: Have NOE NMR experiments been measured for 3j,k? These fast NMR experiments could show some interactions over the space between  H and groups at positions 3 and 3a, this can be useful for the diastereoisomers determination.

10) Mechanism discussion: i) scheme 7: the title should be rewritten instead of “The reaction pathways for 1a3a (or 3’a) transformation”, it should be “Proposed reaction pathways for 1a3a (or 3’a) transformation”; ii) Figure 4: it should clarify TS leading towards 3 and 3´ departing from what species? (OC?); iii) explanation in the main text is not properly detailed, a complete energy calculated profile could be really helpful and clarify the roles of different species and intermediates (for example OC) and both paths (thermodynamic and kinetic), this energy diagram should added (it could be in the SI)

11) Table 4: in the last column, instead of a foot note, the title should be: HPLC yield

12) Table 5: the entries should be correlatively numbered (as in Table 1): 1, 2, 3... The code of the final compounds (8a-j) should be indicated in the last column along with the  yied

13) regarding the effect of selected compounds on the growth of C. vulgaris cells and their enrichment in chlorophyll content: did the authors study the metabolism of the tested compounds? Any metabolized product derived from 3a or 8j were detected? Did the biological samples were evaluated regarding this issue? (this issue could be really important depending on the potential future use of such cells)

14) Characterization:

In 13C NMR characterization of compound 3a, there is one carbon is missing.

In 13C NMR of compound 3c there are extra signals (compounds should present 31 signals, but there are signals corresponding to 35 carbon nuclei)

Please check the NMR characterization (NMR reports)

Comments on the Quality of English Language

The English language is correct (grammar and syntax); however, some sentences are not easy to understand (they were mentioned in comments below)

Author Response

Response to reviewer #3

1) Figure 1: names of alkaloids (or related) should be added into the figure

Response: These compounds are not alkaloids. They are alkaloid-like synthetic chemicals, and they do not have trivial names.

2) Table 1: in the last column, instead of a foot note, the title should be: HPLC yield

Response: We corrected the text.

3) Line 106: is any other by-product identified? Did the authors check consumption of the Schiff base? The monitoring of the reaction was done visually by violet colour disappearance related to the APBTT, can this disappearance of colour be related only to ABPTT decomposition, instead of unselectively reaction with Schiff base?

Response: «Line 106: is any other by-product identified?» - we added comments on it in the text of the manuscript.

«Did the authors check consumption of the Schiff base?» - no, we did not. Consumption of Schiff base can proceed due to side reactions with compounds 1 and/or atmospheric moisture, which makes it just pointless for reaction monitoring.

«The monitoring of the reaction was done visually by violet colour disappearance related to the APBTT, can this disappearance of colour be related only to ABPTT decomposition, instead of unselectively reaction with Schiff base?» - dark violet colour is a characteristic of APBTTs; its disappearance is a marker of APBTT consumption that can proceed both due to reaction of APBTT with a Schiff base with a formation of compounds 3 and due to other (side) reactions of APBTT.

4) Table 2: i) the entries should be correlatively numbered (as in Table 1): 1, 2, 3…13. The code of the final compounds should be indicated in the last column along with the yied; ii) the Schiff bases that were not successful should also be incorporated in the table (at least as footnote, see comment N° 7); iii) n/d abbreviation should be indicated as a table footnote

Response: i) We prefer numbering of the entries by compounds names. This is clearer for the reader and does not consume the page space. ii) We think that incorporation of Schiff bases that were not successful in the table would be illogical and misleading for the reader. iii) We added this footnote.

5) Lines 151-156: “For these reasons, we replaced chloroform with benzene in these reactions. As a result, we noticed a decreasing trend in the HPLC yields of products 3 (Conditions B, Table 2), which correlated with the optimization data for the test reaction of APBTT 1a with Schiff base 2a (Table 1). Moreover, products 3 were easily isolated and purified by simple crystallization directly from the reaction mixtures (in scale of 298 μmol, benzene).”

This analysis paragraph should be re-written: i) “…we noticed a decreasing tendency in the HPLC yields of products 3 for most of the substrates evaluated (compared conditions in entries xxx, table 2)…” ii) Moreover should be changed by “however”.

Did the authors try or evaluate the possibility of non-polar solvent with better properties regarding environment issues (such as cymene)

Response: The text was changed. We did not tried the possibility of non-polar solvent with better properties regarding environment issues (such as cymene). These chemicals are not widespread in laboratories yet, and when we performed this study we just did not take this into account. We believe that investigations of solvents with better properties regarding environment issues on this and similar reactions can become an issue for a separate publication.

6) Regarding the issue discussed above: is the yield measured by HPLC (table 2) representative of the efficacy of the reaction? Could it be possible that part of the product is already precipitated in benzene and, therefore, less product is detected, which would mean that in both tested solvents the formation of the product is comparable, but the isolated yields (because of possible precipitation-crystallization) are better for benzene?

Response: Before the HPLC analysis, all reaction mixtures were homogenised by adding to them additional solvents in order to dissolve the formed precipitates. Thus, the yields measured by HPLC are representative of the efficacy of the reaction.

7) lines 159-165: the examples that were not successful in the studied transformation should be also added in Table 2 as this is also important information to analysis the scope of the reaction. In this paragraph, “APBTTs 1” should be replaced by “APBTTs 1a-g” (if all of them were evaluated, or to detail which of them were tested). Incorporation of these experiments in Table 2 will help to clarify this issue.

Response: We think that incorporation of Schiff bases that were not successful in the table would be illogical and misleading for the reader. “APBTTs 1” was be replaced by “APBTT 1a”.

8) lines 181-182: “In our scope studies (Table 2), we isolated exclusively (3R*,3aS*,11aR*)-diastereomers for compounds 3a-i,l,m.” In the reactions leading to 3a-i,l,m only, were (3R*,3aS*,11aR*)-diastereomers isolated nor detected? In the HPLC analysis (previous to purification) was any other diastereoisomer detected? Were the crude reaction mixtures evaluated by a different method rather than HPLC? The main text should be clear about this issue: if the other diastereoisomers were not isolated, or, if they were not detected nor identified in the reaction mixtures, if analyzed.

Response: We added explanations to the main manuscript.

9) line 189: Have NOE NMR experiments been measured for 3j,k? These fast NMR experiments could show some interactions over the space between H and groups at positions 3 and 3a, this can be useful for the diastereoisomers determination.

Response: We examined some NOE spectra of these compounds, but we did not succeed to find any useful for configuration determination signals in them.

10) Mechanism discussion: i) scheme 7: the title should be rewritten instead of “The reaction pathways for 1a3a (or 3’a) transformation”, it should be “Proposed reaction pathways for 1a3a (or 3’a) transformation”; ii) Figure 4: it should clarify TS leading towards 3 and 3´ departing from what species? (OC?); iii) explanation in the main text is not properly detailed, a complete energy calculated profile could be really helpful and clarify the roles of different species and intermediates (for example OC) and both paths (thermodynamic and kinetic), this energy diagram should added (it could be in the SI)

Response: i) The text was updated. ii) The text was updated. iii) Energy profile of the reaction was added to the main manuscript.

11) Table 4: in the last column, instead of a foot note, the title should be: HPLC yield

Response: The text was updated.

12) Table 5: the entries should be correlatively numbered (as in Table 1): 1, 2, 3... The code of the final compounds (8a-j) should be indicated in the last column along with the yied

Response: We prefer numbering of the entries by compounds names. This is clearer for the reader and does not consume the page space.

13) regarding the effect of selected compounds on the growth of C. vulgaris cells and their enrichment in chlorophyll content: did the authors study the metabolism of the tested compounds? Any metabolized product derived from 3a or 8j were detected? Did the biological samples were evaluated regarding this issue? (this issue could be really important depending on the potential future use of such cells)

Response: In the present study, we did not investigate the metabolism, because metabolism studies are long, difficult, requiring special staff and laboratory equipment. In the future studies, possibly, it would be interesting to study metabolism of these compounds in algae, but the results of such studies would be published in a separate publication.

14) Characterization:

In 13C NMR characterization of compound 3a, there is one carbon is missing.

In 13C NMR of compound 3c there are extra signals (compounds should present 31 signals, but there are signals corresponding to 35 carbon nuclei)

Please check the NMR characterization (NMR reports)

Response: Compound 3a and some other relative compounds have missing signals of aromatic carbons in NMRs, which is a common situation for sterically hindered compounds (some aromatic signals are broadened and cannot be detected without specific experiments). Compound 3c contains F, thus, the number of lines in 13C is more due to 13C splits on 19F.

Round 2

Reviewer 3 Report

Comments and Suggestions for Authors

All the comments were properly addressed. Regarding comment 14: Please as discussed and explained in authors´ answer “Compound 3a and some other relative compounds have missing signals of aromatic carbons in NMRs, which is a common situation for sterically hindered compounds (some aromatic signals are broadened and cannot be detected without specific experiments). Compound 3c contains F, thus, the number of lines in 13C is more due to 13C splits on 19F.”, for 3a characterization a note indicating that 1 carbon signal could not be detected should be added and for 3c characterization proper description should be detailed, indicating the C- F coupling (multiplicity: d, and the J constants)

Author Response

For 3a,c-g,i,3’j,8d-f,i,j characterization a note indicating that some carbon signals could not be detected was added.

For 3c and 8c characterization proper description was detailed.